

# Patients poorly recognize lesions of concern that are malignant melanomas: is self-screening the correct advice?

Mike Climstein[1,2,3,*], Jeremy Hudson[1,4,*], Michael Stapelberg[1,5], Ian J. Miller[1,5], Nedeljka Rosic[1,6], Paul Coxon[4], James Furness[7] and Joe Walsh[8,9]

[1] Aquatic Based Research, Faculty of Health, Southern Cross University, Bilinga, Qld, Australia
[2] Physical Activity, Lifestyle, Ageing and Wellbeing Faculty Research Group, University of Sydney, Sydney, NSW, Australia
[3] Clinical and Health Services Research Group, Faculty of Health, Southern Cross University, Bilinga, Qld, Australia
[4] North Queensland Skin Centre, Townsville, Qld, Australia
[5] John Flynn Specialist Centre, Tugan, Queensland, Australia
[6] Biomedical Sciences, Faculty of Health, Southern Cross University, Bilinga, Qld, Australia
[7] Water Based Research Unit, Bond University, Robina, Qld, Australia
[8] Sports Science Institute, Sydney, NSW, Australia
[9] AI Consulting Group, Sydney, NSW, Australia
[*] These authors contributed equally to this work.

Corresponding author
Mike Climstein,
michael.climstein@scu.edu.au

## ABSTRACT

**Background.** Australia is known for its outdoor culture, with a large percentage of its population engaging in outdoor recreational activities, aquatic, non-aquatic and outdoor occupational activities. However, these outdoor enthusiasts face increased exposure to ultraviolet radiation (UVR), leading to a higher risk of skin cancer, including malignant melanoma (MM). Over the past 40 years, there has been a significant rise in skin cancer rates in Australia, with two out of three Australians expected to develop some form of skin cancer by age 70. Currently, skin cancer examinations are not endorsed in asymptomatic or low-risk individuals in Australia, with only high-risk individuals recommended to undergo regular skin examinations. Notably, the Melanoma Institute Australia suggests that one-half of patients identify MMs themselves, although this claim appears to be based on limited Australian data which may not reflect contemporary practice. Therefore this study sought to determine the percentage of patients who were able to self-identify MMs as lesions of concern when presenting for a skin cancer examination.

**Methods.** Multi-site, cross-sectional study design incorporating a descriptive survey and total body skin cancer screening, including artificial intelligence by a skin cancer doctor.

**Results.** A total of 260 participants with suspect MM lesions were biopsied, with 83 (31.9%) found to be melanomas. Of the true positive MMs only a small percentage of participants (21.7% specificity) correctly had concerns about the suspect lesion being a MM. These MMs were located primarily on the back (44.4%), shoulder (11.1%) and upper leg (11.1%). There was no significant difference in the size between those participants aware of a MM versus those who were not ($P = 0.824$, 24.6 *vs* 23.4 mm$^2$). Significantly more males identified lesions of concern that were MMs as compared to females ($P = 0.008$, 61.1% *vs* 38.9%, respectively). With regard to true negatives males

and females were similar (52.1% *vs* 47.9%, respectively). With regard to false negatives ($n = 65$), a greater percentage of males than females did not recognize the MM as a lesion of concern (66.2% *vs* 33.8%, respectively). Participants were more likely to correctly identify an invasive MM as opposed to an in situ MM (27.3% *versus* 21.3%).

**Conclusions**. Only a small percentage of participants in this study were able to self-identify either *in situ* or invasive MM as a lesion of concern with a tendency to identify the more advanced, thicker MMs. Given that MM is associated with a high mortality and cost of treatment, particularly when invasive, the inability of lay persons to identify these cancerous lesions will likely lead to delayed treatment and a possible adverse outcome. We believe the current melanoma screening practices in Australian general practice should be revisited to improve patient outcomes with regard to MM. Additionally, prevention campaigns should include images and primary risk factors for MM.

# INTRODUCTION

Australia's culture lends itself to being conducive to outdoor activities due to its sunny, warm and dry climate. For example, the national walking and cycling participation survey (*Bolton, 2023*; *Landis & Koch, 1977*) reported that the majority (89%, approximately 23.7 million) of Australians walk at least ten minutes outside their home on a weekly basis. This survey also reported that approximately three million Australians run and approximately four million Australians cycle on a weekly basis. Australians also have high participation in aquatic activities, with an estimated 3.5 million surfers (*Stark, 2013*; *McCubbing, 2022*), and a near equal number (three million) of swimmers (*SportAus Ausplay, 2019*). Additionally, the Cancer Council's national sun survey estimated that over 2.5 million Australians spend one-half of their working time outdoors (*SunSmart, 2015*).

Although there are significant health benefits associated with these outdoor recreational activities (*BetterHealth Victoria, 2023*; *Centers for Disease Control Prevention, 2022*; *Williams & Thompson, 2013*), there are inherent risks. Compared to people who engage in indoor recreational activities, individuals who participate in outdoor activities, regardless of aquatic, land-based or occupational in nature, are exposed to significantly more ultraviolet radiation (UVR) with UVA and UVB associated with skin cancer (*Snyder et al., 2020*). In fact, UVR is a group 1 carcinogen shown to be a mechanism for the development of skin cancer (*Vainio, Heseltine & Wilbourn, 1994*). Because of Australia's geographic location, Australians face a higher UVR exposure, as assessed by the UVR index (*Cancer Council, 2023b*), than those people living in the United States and Europe (*Umar & Tasduq, 2022*). Moreover, it is commonly known that the ozone layer's shielding function is weaker above Australia, which results in higher UVR levels reaching the Earth's surface in Australia (*Umar & Tasduq, 2022*). It is commonly reported that exposure to UVR can cause precancerous skin lesions which include actinic keratosis (AK) and cancerous lesions including malignant melanomas (MM), keratinocyte carcinomas (KC), and basal cell carcinomas (BCCs) and

squamous cell carcinomas (SCCs) (*Feller et al., 2016*) which have been reported to be the most common (*Olsen, Williams & Whiteman, 2014*).

Over the previous 40 years, Australia has seen a significant rise in the incidence of skin cancer cases (*Wada et al., 2020*) and it is estimated that two out of every three Australians will develop some form of skin cancer before the age of 70 years (*Royal Australian College of General Practitioners, 2023a*). Furthermore, in 2022 over 1,500 Australians died from skin cancer (melanoma and other malignant neoplasms of skin) (*Australian Bureau of Statistics, 2023*).

The standardized rate for MM in the Australian general population increased by 81% between 1982 (27 cases per 100,000) to 2018 (54 cases per 100,000), according to data from the Australian Institute of Health and Welfare (AIHW) (*Australian Institute of Health Welfare, 2023*; *Australian Government Cancer Australia, 2022*). The most recent estimate in 2023 by the AIHW (*Australian Institute of Health Welfare, 2023*) shows a further increase in the standardized rate to 69 cases per 100,000. This represents a further 28% increase in the standardized rate of MM over the past five years in Australia. In fact, Australia leads the world in skin cancer incidence with an overall rank of first of all countries and ranks first for males and females (*World Cancer Research Fund International, 2020*).

There is also a significant economic burden associated with skin cancer in Australia. Gorden and colleagues in their costs of illness analysis for melanoma (*Gordon et al., 2022*) reported the first year costs of a newly identified MM and its treatment was $824 million with individual patient costs ranging from $644 (*in situ* melanoma, excision and/or graft/flap) to $100,725 (stage III/IV, unresectable) MM. Specific to aquatic (surfers and swimmers) and non-aquatic (walkers/runners) outdoor enthusiasts, recent research reported (*Miller et al., 2023*) a higher standardized rate in these Australian cohorts. We recently reported surfers had a standardized rate of MM of 6,481 per 100,000 followed by walkers/runners (4,314 per 100,000) and swimmers (3,333 per 100,000) (*Miller et al., 2023*). When compared to the general Australian population, the odds ratios ranged from 61 in swimmers to 119 in surfers. The Australia national economic burden for 2021 was reported at almost $400 million for newly diagnosed MM (*Gordon et al., 2022*).

*Reyes-Marcelino et al. (2023)* completed a 16-year longitudinal skin cancer screening study (2000 to 2016, inclusive) of patients aged 15 years and older who presented with a skin cancer-related condition. Over 15,600 Australian general practitioners (GPs) participated in the study and identified and managed over 65,000 skin cancers (rate of 47.7 per 1,000 patient encounters). They found the rates of skin cancer ranged from 2.4% with MM to approximately 30.0% with actinic keratosis (a pre-malignant lesion). The rates of skin cancer were the highest in the state of Queensland (71.7 per 1,000 patient encounters) and lowest in Victoria (36.6 per 1,000 patient encounters). *Roseleur et al. (2021)* also conducted an Australian longitudinal (2011 to 2020, inclusive) skin cancer study. During this period, 370 general medical practices completed approximately 68,000 skin screening checks and they reported a higher rate of skin cancer with MM at 12.7%, basal cell carcinoma 43.3% and 38.2% squamous cell carcinoma.

It has been widely reported (*Reyes-Marcelino et al., 2023*; *Del Mar & Lowe, 1997*; *Charles, Knox & Britt, 2006*) that GPs primarily provide the diagnosis and treatment of most skin

cancers. However, there has been an emergence in skin cancer medicine as a primary care sub-specialty (*Youl et al., 2007*). *Pandeya et al. (2023)* reported that 77% of MM are initially managed in primary practice. *Reyes-Marcelino et al. (2023)* demonstrated the burden of skin cancer conditions managed by GPs, and the rates of skin cancer treatment were shown to increase over the course of the longitudinal study, including melanoma.

Despite the extraordinarily high prevalence of MM in Australia (*Miller et al., 2023*; *Reyes-Marcelino et al., 2023*; *Roseleur et al., 2021*), there is no national screening program as there is an overall lack of evidence that it would be effective at reducing rates of mortality. Consequently, Medicare, Australia's publicly funded universal health care insurance scheme, does not fund skin cancer screenings for patients lacking high risk features or overt signs of skin cancer. Signs of skin cancer may follow the "ABCDE" rule or "ugly duckling sign" and includes asymmetry, border irregularity, color variation, lesion size greater than 6.0 mm diameter and evolution (new or progressive changes in the appearance).

The Royal Australian College of General Practitioners (RACGP) publication, the "Red Book", is designed to provide GPs with guidance on preventative care (*Royal Australian College of General Practitioners, 2023c*). The Red Book guidelines for skin cancer screening also state that the screening of asymptomatic patients for non-melanoma and melanoma skin cancer is not recommended due to a lack of evidence identifying that screening these individuals reduces mortality (*New Zealand Guidelines Group Te Ropu Ragangi Tohutohu, 2008*). The risk factors for MM are multifactorial and the criteria for identifying which patients are "high risk" is often limited. According to the Red Book, those patients deemed at high risk (previous history MM or greater than five atypical nevi) should be advised to conduct self-skin examinations every three months (*Royal Australian College of General Practitioners, 2023b*). The Red Book does however, state that those individuals deemed high risk (previous history of MM or greater than five atypical naevi) should undergo a skin examination (with or without photography) every six to 12 months, with the frequency of follow-up based upon the disease stage of the melanoma.

The Chair of the RACGP Specific Interests Group in Dermatology stated that Australia is known throughout the world for having high quality and effective skin cancer screening and detection practices (*Liotta, 2023*). *Youl et al. (2007)* assessed the sensitivity of skin cancer diagnosis in Australian GPs as compared to skin cancer doctors. They reported that the sensitivity of GPs and skin cancer doctors for identifying any type of skin cancer was comparable (0.91 *vs* 0.94, respectively). However, specific to MMs, the skin cancer doctors sensitivity was significantly better than the GPs ($P < 0.01$, 0.60 *vs* 0.29, respectively). Although the study *Youl et al. (2007)* identified that GPs had a high sensitivity for diagnosing non-melanoma skin cancer, GPs were shown to have difficulty distinguishing MM from other common pigmented skin lesions. *Harkemanne et al. (2022)* conducted a study with Australian GPs where they assessed their ability to diagnose MM (threshold of 50% correct identifications), then provided a one hour training session. They reported that the number of GPs that met or exceeded the 50% threshold significantly improved their accuracy following the training session, however, when assessed one year later, the same GPs were unable to maintain the improvement in accuracy. *Harkemanne et al. (2022)*

concluded that further refresher training sessions were necessary to maintain the improved diagnostic skills. It is concerning that the acceptable threshold used in this study was only 50%, however, this seems to align with expectations for non-medical persons based upon existing studies. Additionally, it has been reported (*Devllavalle & Sivesind, 2022*; *Ramji, 2017*) that diagnosing MM by visual inspection can be difficult in people of color, especially where the surrounding skin may either mask or match the color of a MM.

Some studies have compared self-discovered *versus* physician MM. *Xavier et al. (2016)* evaluated self-discovered melanomas in a study involving 211 patients. They reported that 41.7% of their patients were correctly able to self-identify melanoma lesions. Of these patients, the authors also reported that over one-third (36.4%) waited more than six months or longer to see their doctor from when they first observed a change in the pigmented lesion of concern. *Brady et al. (2000)* found 57% of patients were able to self-detect MM and that females were significantly more likely to detect their own MM as compared to males ($P < 0.0001$, 69% *vs* 47%, respectively). They also reported that their female participants were more likely to complete self-examinations, while other studies have reported that females were more knowledgeable about skin cancer (*Miller et al., 1996*; *Bergenmar, Törnberg & Brandberg, 1997*). Both of these studies were completed in specialist cancer centers in Brazil and New York, respectively.

*Watts et al. (2021)* conducted a population based study with approximately 2,500 MM patients in New South Wales, Australia, of which 41.7% self-identified MM. They preferenced the collection of invasive *in situ* lesions. *McPherson et al. (2006)* conducted a similar study involving over 3700 Queensland (Australia) residents, of which 44.0% of the participants self-detected the melanomas. They exclusively collected invasive MM and preferenced lesions with a Breslow thickness of >0.75 mm. *McPherson et al. (2006)* in their paper reported that previous studies have reported the self-detection of melanomas rates ranging from 40 to 74%.

The *Melanoma Institute Australia (2022)* has reported that more than one half of all melanomas are first identified by the patient, although it is unclear which study this percentage stems from *Xavier et al. (2016)*, *Watts et al. (2021)*, *McPherson et al. (2006)*, or if it is anecdotal. Regardless, if approximately one half of melanomas are purported to be self-detected, this would equate to a sensitivity of approximately 0.50, which is almost double the sensitivity observed in the Youl and colleagues' study of GPs (0.29) (*Youl et al., 2007*), who are medically trained to recognize MMs.

Self-identification of MMs is a key part of Australia's National strategy for reducing the impact and burden of MM and is supported alongside successful skin cancer education campaigns such as the slip, slop, slap, seek slide campaign (*Walker et al., 2022*; *Trenerry et al., 2022*). Given that self-identification of MM by patients is not well investigated, we sought to determine the percent of patients presenting for a skin cancer screening who would self-identify a lesion of concern prior to a skin cancer screening that was a MM.

## MATERIALS AND METHODS

### Study design

The Human Research Ethics Committee at Southern Cross University approved this study on May 11th, 2020 (2020/47). All participants in this study initially received a "participant information sheet" and then signed written, informed consent. This multi-site, cross-sectional study began with all participants completing a survey and proceeded to a whole-body, artificial intelligence-assisted skin cancer screening by a skin cancer doctor. The study was conducted in Northeast Queensland (NEQ) and in Southeast Queensland/Northern NSW (GC). The study was promoted internally at both skin cancer clinics as well as through the media, which included radio, television, and online newspapers. In addition, individuals were recruited when attending either of the skin cancer clinics prior to their planned skin check.

### Participants

Males and females aged 18 years and over were invited to participate in this study. Participants were either referred to the skin cancer clinic in their geographic location by their local general practitioner/physician, elected to attend based upon media exposure of the study or had elected to attend the primary care skin cancer clinic on their own as a personal health initiative.

### Questionnaire

The survey followed the protocol previously described by *Miller et al. (2023)*. The survey was divided into four sections: physiological demographics, sun exposure relevant to an activity, techniques for preventing skin cancer, and skin cancer risk and history. As part of the section on techniques to avoid skin cancer, all participants were asked if they had a lesion(s) they were concerned about. All questionnaires were completed in the clinic's waiting area. Questions pertaining to the study or survey questions were answered or clarified by either the skin cancer doctor, or the researcher. Following each participants completion of the survey, the clinician then completed the Fitzpatrick skin type section of the questionnaire. Following completion of the questionnaire, the skin cancer doctor then completed a whole-body skin check examination using artificial intelligence of dermatoscopic images of lesions where the number, type and location of suspected lesions were noted within the participants' file.

### Visual skin cancer screening

Two accredited GPs, both with a special interest in skin cancer working in a primary care skin cancer clinic utilized a commercial, high-resolution digital dermatoscope that included artificial intelligence (AI) (Moleanalyzer-Pro, FotoFinder Systems GmbH, Bad Birnbach, Germany) to conduct all participant's whole-body (scalp, head, neck, torso, arms, palms, thighs, legs, soles, nails, and mucosae) skin checks for skin cancer. The digital dermatoscopy system utilised an attachment (Medicam) which included an integrated LED floodlight and LED micro-illumination (polarized and non-polarized) that allowed up to 40 times magnification of any suspicious lesions. All suspicious lesions thought to be

a melanoma were photographed by the digital dermatoscopic system which automatically generated a predicted AI score for MM. According to the systems manual (*FotoFinder Systems GmbH, in press*), the AI score is based upon comparisons with a database of malignant melanomas images, with the AI score indicating how similar a queried lesion photographed is to a typical malignant skin tumour in the database. The AI score, according to the manufacturer's manual, is merely a statistical value that guides the clinician as to how much focus on the suspected lesion of interest. The AI score is not meant to be a clinical decision-making tool (*FotoFinder Systems GmbH, in press*). According to the manual, the AI predictive classifications ranged from 0.0–0.2 (grey color, uninformative); 0.21–0.49 (yellow color, requiring clarification); and 0.5–1.0 (red color, indicates suspicious, observe with high attention).

## Examination

All participants in this study completed a survey with basic demographic information and consented to full-body skin check. The skin checks were completed using the high-definition, digital dermatoscope which included artificial intelligence (AI) where each participant was systematically examined on all skin surfaces, including the scalp, head, neck, torso, arms, palms, thighs, legs, soles, nails, and mucosae (*Miller et al., 2023*). The dermatoscopic system provides an AI score which based upon the manufacturers recommendations (*FotoFinder Systems GmbH, in press*), an AI score between 0.00 to 0.20 (inclusive) and is defined as unsuspicious. Those AI scores between 0.21 to 0.49 (inclusive) are defined as unsuspicious-requiring clarification. An AI score between 0.50 to 1.00 (inclusive) and is defined as suspicious-observe with high attention.

Prior to each participant undergoing the whole body skin check, each participant was asked if they had a lesion(s) of concern. A range of dermatoscopic algorithms were employed by the study's clinicians, such as the Menzies technique, the ABCDE criteria, the 7-point checklist, the classic pattern analysis, the chaos and hints algorithm, and prediction without pigment (a decision algorithm for non-pigmented skin cancers) (*Rosendahl et al., 2014*). It has been reported that these screening methods, when used by an experienced doctor, significantly increases the diagnostic precision of diagnosing AK, KC and MM (*Rosendahl et al., 2012*). Queried melanomas were captured by the Medicam and an AI score was obtained and recorded in the participant's file. A predictive AI score was obtained on all lesions suspected of being a melanoma. The ground truth (gold standard) was obtained *via* histopathology from a commercial laboratory, which validated any suspected lesions that were obtained *via* a punch biopsy or excised.

## Statistical analysis

We initially determined the normality of our data using kurtosis, skewness, Q-Q plots and Kolmogorov–Smirnov tests (*Mishra et al., 2019*). Analyses of our data included demographics, independent sample $t$-tests and chi-square tests with selected outcome variables. A bivariate correlation was conducted on selected outcome variables. Alpha was set *a priori* at $P < 0.05$ to determine the significance between groups.

Artificial intelligence classification of individual lesions were based upon the manufacturer's three predictive classifications which included unsuspicious, requiring

clarification and suspicious (*FotoFinder Systems GmbH, in press*). The AI diagnostic classification was also based upon the manufacturers score information where an unsuspicious AI score was 0.0 to <0.50 and suspicious was ≥0.5 to ≤1.0 (*FotoFinder Systems GmbH, in press*). An error matrix or confusion matrix, (*Kulkarni, 2020*) was used to report the classifications and calculate sensitivity and specificity (*Parikh et al., 2008*).

The data set was limited to only queried lesions thought to be MM as this was the focus of the study. All statistical analyses were computed using IBM's Statistical Package for Social Sciences (SPSS, Ver. 28.0).

## RESULTS

### Participants' characteristics/demographics

A total of 260 (males $n = 143$, females $n = 117$) participants who attended a skin cancer screening from the combined clinics (GC $n = 172$, NEQ $n = 88$) were identified with a possible melanoma lesion that was biopsied and sent for histopathology (Table 1). The GC had a higher number of participants with the majority (59.3%) being male whereas NEQ had a higher percentage of female participants (53.4%). With regard to age of participants between the two clinics, there was no statistically significant difference between the two clinic sites ($P = 0.543$, GC 54.4 *vs* NEQ 53.2 years). Of all participants in this study, only a small percentage were referred by their GP to participate in this screening study (7.7%). Of those participants referred to the study by their GP, the majority of participants were those seen at the GC clinic *versus* NEQ ($P = 0.015$, 16.3% *vs* 5.7%, respectively).

### Whole-body screening results

The skin cancer specialists identified 260 lesions thought to be a MM, of which 83 (31.9%) were determined to be a MM *via* histopathology (female = 29 (34.9%), males = 54 (65.1%)). These melanomas were primarily located on the back (34.9%), lower leg (13.3%) arms (10.8%) chest (9.6%) and elsewhere on the body ranging from the shoulders and upper leg (7.2%) to the ear (2.4%). Considering the participants identified with a MM, 32% had a previous MM diagnosed and treated. Of all participants with a MM identified in this study, 68% had a family history of MM. Of all participants ($n = 260$), 26.9% had a history of MM, of this percentage, 8.5% had both a history of MM and a MM identified in this study. Additionally, 35% of the participants in our study identified with a MM also had a family history of MM.

Additionally, there were a number of participants ($n = 5$) who had two MM. Of the five participants, one participant was aware of one of the MM located on the thigh. However, this same participant was not aware of the other MM which was on the neck. Three participants out of five were unaware of two MM *in situ* located on their bodies. The locations of the MM in these participants were the abdomen and neck ($n = 1$ each), back and shoulder ($n = 1$) and the back and neck ($n = 1$ each). The final participant ($n = 1$) with multiple MM was not aware of either malignancy, which were both located on the back. One of these lesions was a MM *in situ* and the other lesion was an invasive MM.

The mean size of all confirmed MM ($n = 83$) was 24.4 mm$^2$ (range 2.5 to 100 mm$^2$, ±SD 21.1 mm$^2$) The lesions identified in the GC clinic were significantly smaller than

**Table 1  Participant's demographics, values are mean/median (±SD), number or percent (%), and 95% confidence intervals (95% CI).**

| Parameter | Group (n = 260) | GC (n = 172) | NEQ (n = 88) | P-value (GC vs NEQ) |
|---|---|---|---|---|
| Age (years) | 53.96 (15.2) | 54.37 (14.2) | 53.15 (17.1) | 0.543 |
| | [52.1–55.8] | [52.2–56.5] | [49.5–56.8] | |
| Gender (n, %) | | | | |
| Males | 143 (55.0) | 102 (59.3) | 41 (46.6) | 0.051 |
| Females | 117 (45.0) | 70 (40.7) | 47 (53.4) | |
| Lesion of concern: Yes (n, %) | 50 (19.2%) | 33 (19.2%) | 17 (19.3%) | 0.552 |
| | [0.14–.024] | [0.13–0.25] | [0.11–0.28] | |
| Males | 24 (16.8) | 20 (19.6) | 4 (9.8) | |
| Females | 26 (22.2) | 13 (18.6) | 13 (27.7) | |
| All suspect lesion sizes (mm$^2$) | 17.7 (19.0) | 15.9 (15.2) | 21.7 (24.4) | $P = 0.019$ |
| | [15.5–20.2] | [13.9–18.3] | [16.5–26.9] | |
| Melanoma lesion size (mm$^2$) | 24.4 (21.1) | 20.1 (16.0) | 40.5 (29.7) | $P < 0.001$ |
| | [19.7–29.0] | [16.1–24.1] | [25.2–55.8] | |
| AI score (all) | .49 (.35) | .52 (.32) | .43 (.39) | 0.026 |
| | [0.45–0.54] | [0.47–0.56] | [0.35–0.52] | |
| AI score (melanomas only) | .60 (.33) | .60 (.04) | .60 (.37) | 0.985 |
| | [0.53–0.67] | [0.53–0.68] | [0.41–0.80] | |
| AI Score classification all lesions (n, %) | | | | 0.005 |
| Unsuspicious | 91 (35.0) | 49 (28.5%) | 42 (47.7%) | |
| | [0.29–0.41] | [0.22–0.35] | [0.37–0.58] | |
| Requiring clarification | 47 (18.1) | 37 (21.5%) | 10 (11.4%) | |
| | [0.13–0.23] | [0.15–0.28] | [0.05–0.18] | |
| Suspicious | 122 (46.9) | 86 (50.0%) | 36 (40.9%) | |
| | [0.41–0.53] | [0.43–0.57] | [0.31–0.51] | |
| AI diagnostic category (n,%) | | | | 0.104 |
| Positive ≥ 0.50 | 122 (28.6) | 86 (50.0%) | 36 (40.9%) | |
| | [0.41–0.53] | [0.43–0.57] | [0.31–0.51] | |
| Negative ≤ 0.49 | 138 (32.3) | 86 (50.0%) | 52 (59.1%) | |
| | [0.41–0.59] | [0.43–0.57] | [0.49–0.69] | |
| Melanoma Histopathology (n,%) | | | | 0.01 |
| Positive | 83 (31.9) | 66 (38.4%) | 17 (19.3%) | |
| | [0.26–0.38] | [0.31–0.46] | [0.11–0.28] | |
| Negative | 177 (68.1) | 106 (61.6%) | 71 (80.7%) | |
| | [0.62–0.73] | [0.54–0.69] | [0.72–0.89] | |

**Notes.**
GC, Southeast Queensland/Northern NSW; NEQ, Northeast Queensland.

those identified at NRQ ($P < 0.001$, 20.1 *vs* 40.5 mm$^2$). With regard to gender, the majority (65.1%) of MM were identified in males who had significantly larger lesions ($P = 0.046$, 27.7 *vs* 14.4 mm$^2$) than females. The mean AI score for all MM was 0.60 (±0.32), with no statistically significant difference identified ($P = 0.985$) between the GC and NRQ clinics (0.60 *vs* 0.60, respectfully). Nearly one half of lesions sent for histopathology analysis were categorized as "suspicious" (46.9%) by the Vexia system. Of the lesions that were confirmed to be MM, the Vexia system correctly identified 51 MM (61.4%) as "suspicious" while missing to identify as "suspicious" 32 out of 83 (38.6%) confirmed MM . There was

**Table 2 Participant's lesion of concern awareness and histopathology.** Values represent the number, percent (%) and [95% confidence interval].

| | Histopathology | |
| --- | --- | --- |
| | Positive (Confirmation of melanoma) | Negative |
| **Patient aware** | 18 subjects (21.7%) [12.8% to 30.6%] *7 females (38.9%) [16.4% to 61.4%] 11 males (61.1%) [38.6% to 83.6%]* | 31 subjects (17.5%) [11.9% to 23.1%] *18 females (58.1%) [40.7% to 75.4%] 13 males (41.9%) [24.6% to 59.3%]* |
| **Patient not aware** | 65 subjects (78.3%) [69.4% to 87.2%] *22 females (33.8%) [22.3% to 45.3%] 43 males (66.2%) [54.7% to 77.7%]* | 146 subjects (82.5%) [76.9% to 88.1%] *70 females (48.0%) [39.8% to 56.0%] 76 males (52.0%) [44.0% to 60.2%]* |
| **Total** | 83 subjects (100.0%) *29 females (34.9%) 54 males (65.1%)* | 177 subjects (100.0%) *88 females (49.7%) 89 males (50.3%)* |

no statistically significant correlation identified between the size of the lesions and the AI scores ($P = 0.237$) or the participants age ($P = 0.120$).

Out of the 83 participants with confirmed cases of MM, only 18 participants (21.7% participant sensitivity) were able to recognize a lesion of concern (Table 2). The MM self-recognized by participants were primarily located on the back (44.4%), shoulder (11.1%), upper leg (11.1%) with the remaining lesions equally distributed (5.6% each) on the abdomen, lower leg, arm, chest, face and neck. Most participants ($n = 65$, false negative rate 78.3%) who had a MM were unaware of the lesion. These lesions were primarily located on the back (32.8%), lower leg (15.6%) with the remainder distributed from the arms and chest (10.9% each) to the ear and face (3.1% each). Of the confirmed MMs on the back, nearly one third (29.6%) of participants had awareness, however the majority (70.4%) of participants were unaware. There was no statistically significant difference in age identified between those individuals who were aware of having a MM *versus* those participants who were not aware ($P = 0.164$, 59.0 *vs* 55.8 yrs). There was also no statistically significant difference identified in regard to the size of the lesion between those who were aware of a MM *versus* those who were not (P = .824, 24.6 *vs* 23.4 mm$^2$) (Fig. 1). Even in larger lesions (*i.e.*, >60 mm$^2$), participants were unaware the lesion was a MM.

Regarding MM being *in situ* or invasive, there was no statistically significant difference identified in the age distribution ($P = 0.473$) between the patients with confirmed MM based on category of age (younger than 40 *vs* 40 years and over). The majority (91.7%) of MMs were *in situ* in individuals aged 40 and over, while most invasive MMs were also in the 40 years and over age group (86.4%).

Following investigation *via* histopathology of lesions suspicious of malignancy, the majority of participants ($n = 177$) did not have a MM; However, 31 participants (17.5%) had identified a lesion of concern that was investigated *via* histopathology, which was then classified as either not a MM or a skin cancer (false positive rate).

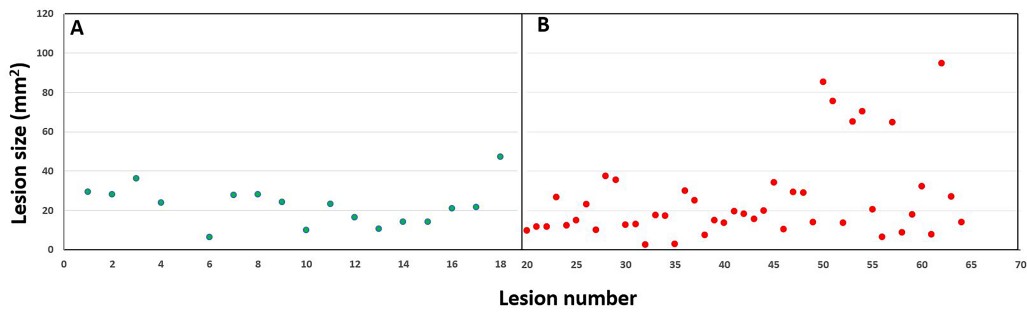

**Figure 1  Lesion size in the participants who suspected they had a melanoma (A, green circles) and those participants who were unaware the lesion was a melanoma (B, red circles).**

Of the 83 MM, the majority were *in situ* (72.3%) *versus* invasive (27.7%). The majority of *in situ* and invasive MMs found in males (66.7% *vs* 63.6%, respectively). There was also no significant difference in size ($P = 0.341$) between the *in situ* MM *versus* the invasive MM, however, the invasive MM were slightly larger than the *in situ* MM lesions (+9.2%, 23.8 *vs* 26.0 mm$^2$, respectively). Additionally, participants were more likely to correctly identify an invasive MM as opposed to an *in situ* MM (27.3% *versus* 21.3%).

# DISCUSSION

Skin cancer is the most common cancer in Australia, which has the highest skin cancer rates in the world (*Bray et al., 2018*), and represents a significant health concern, particularly with regard to MM where early detection and treatment is essential for optimal patient outcomes. In Australia, the majority of skin cancer is diagnosed and managed in general practice (*Reyes-Marcelino et al., 2023*). In this study we identified the percentage of patients who presented for a total body skin check who suspected a lesion of concern was a MM. This study contributes to the limited literature on the ability of patients to self-identify lesions of concern that are MMs when presenting for a skin check in Australian general practice. A total of 260 participants participated in this study who had lesions suspect of being MM and subsequently underwent biopsies on those lesions which were sent for histopathological analysis. Of the 260 suspect lesions, 83 were histopathology confirmed MMs (31.9%). Only a small percentage (21.7%) of participants presented with lesions of concern they thought to be a MM, that was later confirmed (*in situ* or malignant). A much larger percentage (78.3%) of the participants were not aware that they presented with a skin lesion that was a MM.

This is not the first study which has investigated a patient's ability to identify MMs. *Xavier et al. (2016)* investigated self-discovered melanomas in a similar size cohort ($n = 211$) of Brazilian patients in a specialized oncology center. They reported a higher, almost double the percentage of patients who correctly identified lesions that were MMs as compared to our study (41.7% *vs* 21.7%, respectively). These authors also questioned their participants on whether they knew melanomas were a serious skin cancer, of which only one-third (36.4%) were aware. They also only reported a smaller amount of *in situ* lesions compared

to our study (23 *vs* 72.3%). *Brady et al. (2000)* also performed in a specialist oncology unit, though in New York, and reported 57% of self-identified lesions. They reported an even lower fraction of in-situ lesions at 10.4%

The population based surveys of *McPherson et al. (2006)* and *Watts et al. (2021)* conducted the largest study of patients being able to self-identify lesions as melanoma, with a similar percentage (44% and 47%, respectively) being able to detect the MM lesions as seen in *Xavier et al. (2016)*. Both of these studies were biased to more advanced and invasive MM. *McPherson et al. (2006)* excluded MM that were *in situ* from their data set whilst *Watts et al. (2021)* preferred invasive lesions and only had 12% of MM lesions that were *in situ*.

All four studies referenced in the above paragraph have reported significantly higher rates of self-reported MM. This could be explained by our study being performed in primary care skin cancer clinics where no referral was necessary, and MM is potentially identified earlier. We reported much higher rates of in-situ lesions at 72.3%. So less advanced melanoma that are generally more subtle with less clues to diagnosis. These malignancies would be more difficult to self-identify and require the expertise of the physician and their equipment. Given that 66.1% of melanomas in Queensland in 2018 were *in situ* it could be suggested that our cohort of lesions are more representative of what is truly seen in the population (*Olsen et al., 2022*). Additionally, in all four studies the MM were thinner in Breslow thickness when identified by the physician. Given the improved prognosis with thinner Breslow thickness this supports the notion of patient screening by physicians.

The Cancer Council's "slip slop slap" skin cancer health (*Walker et al., 2022*) campaign launched in 1980-1981 which was updated as part of the SunSmart campaign to the "slip, slop, slap, seek, slide" slogan. Neither campaign included any image or description of what a melanoma looks like or risk factors for MM. In 1988, the Cancer Council Victoria launched the SunSmart program with an Australia wide rollout, with follow-ups in 1994 for primary schools and 1996 for early childhood. The SunSmart program today focuses upon the risks and benefits of sun exposure, the importance of early screening and early detection and other areas related to skin cancer prevention. This program encourages Australians to be familiar with their skin and to consult a GP if they notice any change in a lesion (shape, color, size) or development of a new lesion.

In 2010 the Cancer Council conducted a campaign termed "no tan is worth dying for" (*Cancer Council, 2010*), which also contained no images nor risk factors for MM. The Cancer Council website (*Cancer Council, 2023a*) does however, provide consumer information on MM signs, symptoms, mechanisms, diagnosis and treatments. This website also contains a link to the ABCD (asymmetry, border, color, diameter) of MM (and other skin cancers) detection. Given the paucity of information available to the public, it may be considered unreasonable that the public has been adequately informed to allow Australians to self-identify a MM. This may explain the focus of campaigns on MM prevention as opposed to MM identification. Given the large number of MMs identified on the backs of our participants, it is highly improbable that self-screening would identify these lesions. Therefore, we recommend spouse or partner screening, which is in agreement with *Robinson (2020)* who reported on a study involving MM survivors. Dr Robinson stated that partners

should conduct skin checks as the partner is able to see areas of the body that are at-risk, which the person themselves cannot see easily, this includes the ears, back, neck and scalp.

It has been reported that GPs have difficulty in making an accurate diagnosis of MM (*Bird, 2004*). However, Australian GPs have been found to be comparable to international dermatologists in benign to malignant ratio of biopsies. *Coetzer-Botha et al. (2023)* evaluated the diagnostic accuracy of Australasian GPs, they reported that the number needed to treat MM (5.73) was lower than the average number needed to treat for dermatologists (9.6) which was identified in a meta-analysis (including 36 studies) (*Petty et al., 2020*).

It is well documented that MMs can be difficult to detect. For example *Rose, Argenziano & Marghoob (2010)* reported that some MMs evade the ABCD rule and mimic non-melanocytic lesions. The authors further added that dermatoscopy has significantly improved to increase the diagnostic accuracy *versus* naked-eye examinations. However, some MMs, even with the use of dermatoscopy, remain challenging to detect due to only subtle clues. It is therefore understandable that GPs have a lower sensitivity in being able to diagnose MMs as compared to non-melanoma skin cancers. *Skvara et al. (2005)* investigated the limitation in the recognition of MM. They concluded the 7-point checklist, pattern analysis, and the dermatoscopy ABCD rule were unable to provide a diagnosis of MM with sufficient accuracy. The authors commented that no single dermatoscopic characteristic or combination of characteristics in the suspect lesions ($n = 325$) could accurately distinguish between melanocytic nevi and MM. Therefore, in clinical practice, GPs and skin cancer doctors do not use one isolated algorithm, checklist or rule. Rather, GPs and skin cancer doctors use a combination of these strategies as well as pattern recognition, experience, and clinical acumen, further highlighting the difficulty of a laymen (public citizen) detecting MMs.

The majority of our non-MM lesions were identified as nevi (low and high grade atypia). British Columbia Cancer (*BC Cancer, 2023*) has reported that early melanoma and a severely atypical nevus are nearly identical with both of these lesions frequently exhibiting significant irregularity in both color/shade and edge. Atypical nevi are clinically significant as it has been reported (*Crutcher & Cohen, 1990*) that individuals with atypical nevi have a six percent to 10-fold lifetime risk of MM (*Gandini et al., 2005*). If the person has a history of MM and multiple atypical nevi, the risk of further MM is increased to 50 percent (*Crutcher & Cohen, 1990*).

*Whiteman et al. (2022)* in Australia conducted a study from a subset of the large QSkin Sun and Health study (*Olsen et al., 2012*) which had 43,762 participants (male and female) who were randomly selected from a register. *Whiteman et al. (2022)* found that those individuals who participated in a skin cancer screening had higher rates of MM (and other skin cancers) than those who did not. The authors stated that individuals who undergo a screening will have a 29% higher rate of MM detected. It was estimated that one-third of *in situ* MM may not escalate to harm a patient. However, presently there is no recognized approach to differentiate which of these lesions will and will not eventually be harmful in a clinical setting. Overall, the majority of MM were not identified by the participants in this study. It is reasonable to speculate that many of the participants would have never

identified the MM until it was possibly too late, leading to a poor prognosis and patient outcome.

## Study limitations and strengths

The authors note several strengths and limitations of this study. Whilst there was a relatively smaller sample size than some other studies such as *McPherson et al. (2006)* from Queensland (Australia), our study implemented histopathological examination for all collected biopsies and excised tissues at a professional laboratory to validate MM cases, with histopathology considered the definitive method for diagnosing MM. Our findings are indeed similar to *Xavier et al. (2016)* who investigated self-discovered MM in 211 Brazilian patients. One addition that could be made to future research would be to question if the lesion of concern was noticed by the participant or someone else such as a partner, friend or relative. Also, it would be a good addition to question why the lesion was of concern to the participant. For example, a change in color, size or appearance as this would aid in highlighting what participants are valuing as a key factor for lesions of concern. We did not track data on patients who decided not to participate, nor the reasoning on why they declined to participate, which may be considered a limitation of this study. The overwhelming majority of our participants were classified as Fitzpatric skin type 1, 2 and 3. Therefore we are limited in the wide generalizability of our findings to dark-skinned Australians and people of color. The strength of our study was our inclusion of histopathology analysis of lesions under suspicion of melanoma in primary care clinics. All excised tissues were sent to a commercial laboratory to confirm if the suspect lesion was cancerous or not . Histological assessment remains the gold standard for melanoma diagnosis (*Melanoma Research Alliance, 2024*). Additionally, all of our skin cancer doctors are highly experienced (five years or greater identifying skin cancers) and had the adjunctive use of a high definition dermatoscope that included artificial intelligence to assist them in their identification of MM.

The interpretation of the 95% confidence intervals given in this manuscript for Table 2 should be considered in the context of some assumptions which were made in their calculation. Firstly, it's presupposed that the sample from which the intervals were derived accurately represents the broader population. Deviations from this, due to a non-random sample or specific subgroups being over- or under-represented, can lead to intervals that do not fully reflect the population parameters. For example, an issue discussed in the preceding paragraph which affects this could be the prevalence of different Fitzpatrick skin types. Moreover, the normal approximation assumes that the data follow a binomial distribution, which can be approximated by a normal distribution when the sample size is large enough. The assumption of normality in our sampling distribution, justified by sufficiently large sample and population sizes, may not be entirely robust, although it is a reasonable approximation given the sample and population sizes are both deemed sufficiently large (quantitatively assessed using the magnitudes of np and n(1-p), where n is the sample size and p is the sample proportion).

The independence of sample observations is also important; if this condition is violated, for example, by the inclusion of related individuals or clusters within the data, the reliability

of the confidence intervals may be compromised. In addition, these intervals presume a stable population proportion during the study period, which may not hold true if there are underlying temporal trends or interventions affecting the population. The intervals were constructed under the presumption of a fixed sample size, determined prior to data collection, and not based on the data itself. The use of adaptive or sequential sampling methods would necessitate alternative analytical techniques and could influence the interpretation of the intervals, this could be an avenue to investigate in future research.

## CONCLUSIONS

Exposure to UVR is documented as a group 1 carcinogen and causal for the development of skin cancers, both non-melanoma and MM. Early detection and treatment are critical for survival from MM as it is strongly associated with the depth of invasion as deeper MM are more likely to metastasize making successful treatment difficult. Therefore early detection is critical for optimal patient outcomes and survival.

A number of organizations have stated that one-half (Cancer Council) to the majority (Melanoma Institute Australia) of MM are self-detected by patients. The literature reports self-detection rates ranging from 40 to 75% however, in our study we found less than one quarter of participants (21.7% sensitivity) were able to self-identify a lesion of concern as a MM. Therefore, a greater emphasis is required on routine screening and skin cancer checks in asymptomatic individuals. Current screening guidelines by the RACGP and Medicare do not fund screening in low risk/asymptomatic individuals due to a lack of evidence that low risk screening reduces mortality. Further, wide screening may be viewed as over-servicing and over diagnosis with associated potential harms.

The majority of MM found in our participants were *in situ*. This is consistent with the literature that physician identified MM are found earlier and thinner than patient identified MM. Given that invasive MM are more difficult to treat and associated with poorer patient outcomes (*i.e.,* potentially lethal) and higher costs, we question the validity of self-diagnosis of MM by the lay public. Our findings support a review of current skin cancer screenings practices as well as contemporary skin cancer campaigns promoting self-diagnosis. It is well reported in the literature that GPs have difficulty in diagnosing MM as they may not fit the ABCD rule, checklist, or patterns and therefore may be mistaken for atypical naevus. Therefore, we recommend the current screening guidelines for MM be revisited in an effort to promote early detection, improved patient outcomes and reduce the economic burden. The improved screening guidelines would be especially important for rural and remote regions of Australia.

## ACKNOWLEDGEMENTS

We would like to express our appreciation for all the participants who took part in our study. We would also like to acknowledge Johnson & Johnson for supporting our stream of skin cancer research which enabled this study to be conducted. We would also like to extend our sincere thanks to Professor Pat O'Shea, friend and mentor, for instilling a passion for research; you are sincerely missed but not forgotten.

### Funding

The authors received funding from Johnson and Johnson for the purchase of the high-resolution digital dermatoscope with artificial intelligence which enabled that aspect of the study to be conducted. The funders had no role in study design, data collection and analysis, decision to publish, or preparation of the manuscript.

### Grant Disclosures

The following grant information was disclosed by the authors:
Johnson and Johnson for the purchase of the high-resolution digital dermatoscope with artificial intelligence.

### Competing Interests

A/Prof Mike Climstein is a Section Editor for PeerJ (Sports Medicine and Rehabilitation); A/Prof Michael Stapelberg and Ian Miller are employed at John Flynn Hospital Specialist Centre; A/Prof Jeremy Hudson and Dr Paul Coxon are employed at North Queensland Skin Centre; Joe Walsh in employed at Sports Science Institute and AI Consulting Group.

### Author Contributions

- Mike Climstein conceived and designed the experiments, analyzed the data, prepared figures and/or tables, authored or reviewed drafts of the article, and approved the final draft.
- Jeremy Hudson conceived and designed the experiments, performed the experiments, authored or reviewed drafts of the article, and approved the final draft.
- Michael Stapelberg conceived and designed the experiments, performed the experiments, authored or reviewed drafts of the article, and approved the final draft.
- Ian J. Miller performed the experiments, analyzed the data, prepared figures and/or tables, authored or reviewed drafts of the article, and approved the final draft.
- Nedeljka Rosic conceived and designed the experiments, analyzed the data, prepared figures and/or tables, authored or reviewed drafts of the article, and approved the final draft.
- Paul Coxon performed the experiments, authored or reviewed drafts of the article, and approved the final draft.
- James Furness analyzed the data, prepared figures and/or tables, authored or reviewed drafts of the article, and approved the final draft.
- Joe Walsh analyzed the data, prepared figures and/or tables, authored or reviewed drafts of the article, and approved the final draft.

### Human Ethics

The following information was supplied relating to ethical approvals (i.e., approving body and any reference numbers):

This study was approved by the Southern Cross University Human Research Ethics committee: 11th May 2020 (2020/047).

## Data Availability

Raw data are available in the Supplemental Files.

## Supplemental Information

Supplemental information for this article can be found online at http://dx.doi.org/10.7717/peerj.17674#supplemental-information.

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
