# Peer review of "Patients poorly recognize lesions of concern that are malignant melanomas: is self-screening the correct advice?"

_PeerJ, doi:10.7717/peerj.17674_

## Round 0.1 · original submission · Major Revisions

I believe that the work is highly relevant to the field and provides an important outlook on the problem of self-detection of melanoma lesions. However, the reviewers suggested several improvements, and importantly, additional controls and reporting checkpoints that are very important to confirm the validity of the findings.

Please pay special attention to Reviewers 3 & 5.

·

Basic reporting

No comment

Experimental design

No comment

Validity of the findings

No comment

Additional comments

Conclusions: Only a small percentage of participants in this study were able to self-identify either in situ or invasive MM as a lesion of concern.

I would dare to emphasize more the Conclusion: Only a small percentage of participants in this study were able to self-identify any MM as a lesion of concern.

To my mind makes it shorter and crisp for the lay reader (policy makers).

·

Basic reporting

Not applicable

Experimental design

Not applicable

Validity of the findings

Not applicable

Additional comments

As the principal doctor and owner of a skin cancer clinic in rural South Australia it has been my observation that the vast majority of melanomas that I diagnose are not something that the patient was aware of. Certainly most melanomas are not brought to my attention by the patient. I am pleased that the research in this paper supports this notion which I believe therefore supports the case for skin cancer screening checks in low risk as well as asymptomatic patients.

Reviewer 3 ·

Basic reporting

No comment.

Experimental design

No comment.

Validity of the findings

No comment.

Additional comments

This is solid work and well presented. Thanks.
Specific Comments
1] Abstract, line 32 - Please capitalize "A" at the start of the sentence.
2] Abstract, Lines 37-39 - I do not see this data (i.e., ...more males identified lesions of concern ... to females) in the results section of the manuscript (my apologies if I missed it).
3] Introduction, lines 89-93 - Please provide references for these statements.
4] Introduction, line 107 - Should not the words "does not" be inserted between "scheme," and "fund"?
5] Introduction, lines 108-111 - This sentence about signs of skin cancer seems out of place per the context of what you are talking about. Please take a look.
6] Introduction, lines 150-152 - Do you have any speculation as to why Brady et al.'s results were opposite of what you found (i.e., difference in detection rates between males and females)?
7] Results, line 236 - Please place "of" between "total" and "236". Thanks.
8] Results - See comment #2.
9] Discussion, line 296 - Please place "and" between "(42)," and "represents". Thanks.
10] Discussion, lines 315 and 316 - Any speculation as to why these researchers reported values double yours?
11] Discussion, lines 340-352 - You open this paragraph noting that MMs can be difficult to detect. As a reader, I find this very relevant to your findings. Per that, you may consider closing out your paragraph with some additional wording (e.g., lines 351 and 352 - Rather, they use a combination of these strategies as well as pattern recognition, experience, and clinical acumen, further highlighting the difficulty of a laymen (public citizen) detecting MMs). Please consider.
12] Discussion, lines 360-367 - I am confused by the logic here. Or is something missing? That is, of course those who participate in screening will have higher cancer rates than those who did not. How would those who did not participate know they had cancer? And, of course those who undergo screening will have a higher rate of MM detected. Again, is something missing here?
13] Discussion, lines 366 and 367 - Your speculation here is highly ambiguous. Perhaps saying ... It is reasonable to speculate that many of the participants would never identify the MM until it was too late. (?)
14] Discussion, lines 368-373 - Please consider deleting this. I do not understand the relevance in context to how you have crafted your discussion. Thanks.
15] Limitations & Strengths, lines 386-388 - Did the work you cite use this technology?
16] Table 2 - In my copy, the middle column, bottom sector, is blackened out making the enclosed data almost impossible to read. Please fix. Thanks.

Reviewer 4 ·

Basic reporting

The manuscript was well written and provided a clear need for the study. Additionally, background material was presented that was pertinent to the investigation. Results from the current study were clearly presented and tables and figures added value to the written results. The results addressed the hypothesis of the study and therefore provided evidence for decisions.

Experimental design

The study was well designed and furthered previous research with the tools used to identify MMs. The methods were well reported and had clarity. Overall, the investigation displayed rigor and depth and addressed the well defined research question.

Validity of the findings

The data and results were sound and provided meaningful information relative to the need for enhanced education in skin cancer screening. The authors may consider providing a bit more speculation in the discussion as to why their results and those of previous research (references 31, 33) were so numerically different.

The limitations and strengths of the study were well thought out and clearly presented

The conclusions were important to the health of Australian population and emphasized the need for better education on skin cancer self screening. Overall this is an important paper and sheds light on a need of the Australian society.

Reviewer 5 ·

Basic reporting

The structure of the report is sound and the communication is good. There are some minor copy edits to be made and some language with the statistics that would need to be modified to address the issues mentioned below.

There might be a gap in the paper. This might be appropriate to mention here. There is some literature that indicates that skin color matters in regard to diagnosis and risks, which might be addressed at some point(s) in the paper.

There may be better papers to cite at line 72. The cited paper has a focus on treatment rates rather than reviewing the body of work on the association between UVR and precancerous skin lesions. Line 214-215 is not supported with a reference.

Experimental design

The study design is sufficient for the purpose. There were some instances where a full evaluation was difficult because some information or explanation wasn't explicit.

Regarding the survey design/methods, the authors cite the Miller paper, which then cites Climstein et al. I was not able to quickly get a sense if there was more information regarding the recruitment process. Was everyone entering the clinics approached and invited? Was it just a small number of clinics? Did the media campaign bring many people in from the community who were not already attending these clinics or would have attended? The sample is a convenience sample and a volunteer sample mixture, I think. The STROBE checklist says page 7 contains info that I didn't see (maybe I overlooked it). How many people were approached and declined to participate? I'll mention this below. The results are important and solid, but the generalizations that can be made are strongly limited by the method of participant ascertainment. How representative of the local population was the sample (and GPs)? It would benefit the reader to be able to gauge that.

Regarding design: Was each person limited to just one lesion? What about when the lesion of concern was different than the one biopsied? I saw in the data file there were only 260 rows of data (1 per participant?). This seems worth confirming.

Line 231: State who/what determined what was thought to be melanoma explicitly. Knowing this can change the understanding of the findings.

Validity of the findings

Accepting the statistical null hypothesis should be avoided. In many places, equality is accepted in the way the authors word their statements when the P-value is large. This should be avoided. It might also be useful to consider the CIs of the differences to gauge how large the true differences might be (but this assumes statistical validity, which is based on random sampling. This is a limitation that should be integrated into the conclusions.)

It seems unclear if the target of the analysis is useful (and thus misleading) or if it could be bolstered by additional analyses. The analyses seem to be trying to determine if people can detect melanoma. Perhaps this should be assessed in parallel with determining if patients can determine which lesions should be biopsied. To make this point clearer, it would be good to expand Table 2 with clinician judgment and AI categorization versus the pathology findings.

Generalization of the findings appears to be very constrained due to the ascertainment method and the participants' representativeness on a global scale. This is hard to gauge without more information, and the authors guide the reader through a rationale for how far we can generalize. Again, I come back to wondering if this only applies to light-skinned or dark-skinned Australians.

Additional comments

Table 2 could benefit the reader by including the percentage by sex in addition to the percentages for the total (include CIs).

I can't evaluate Figure 1. I don't understand the X axis unless its just an arbitrary index. If that's the case, maybe a dot density plot would be better.

Please include CIs with your point statistics. It would be beneficial to add the test statistic and degrees of freedom with the P-values in the body of the text. Seeing the df's would aid in detecting issues.

I hope the comments are useful.

---

## Round 0.2 · Minor Revisions

The manuscript has significantly improved and the reviewers require only the reporting updates that do not require intense re-writing or re-framing of the findings. Note the confidence intervals, x-axis units of measurement, and density plot suggestions. Also, please, address the suggestion of one of the reviewers to run the non-independence modeling.

Reviewer 3 ·

Basic reporting

No comment.

Experimental design

No comment.

Validity of the findings

No comment.

Additional comments

Thanks for your attentiveness to the review process.

Reviewer 5 ·

Basic reporting

The article has improved in clarity and now addresses several gaps identified by various reviewers.

Figure 1 should be replaced. It appears that the x-axis is an arbitrary value. For the purpose suggested in the response, it would be more advantageous to present a dot density plot by whether or not an MM was identified as a lesion of concern (perhaps that set should be limited to those people with MM?)

Experimental design

The description is now clarified in the body of the text and in the responses.

Validity of the findings

Some of the issues with the non-independence of observations (i.e., multiple observations from the same person in some cases) were addressed with a new section in the limitations. The impact may be small because there are so few multiple observations. It would be best to have modeled this non-independence directly. It violates the assumptions of the reported statistical models, and this is recognized. If the paper reports the original models, it is important for the report to quantify how much (and in which direction) this violation might be impacting the reported statistics. Are the findings different if the violation is addressed?

I do not understand why the CIs for proportions are not being reported. There is a substantial literature evaluating the fine differences among many different stats for proportions. There are tools available for gauging the variance of the estimates of proportions. They should be reported in most instances.

---

## Round 0.3 · accepted · Accept

Thank you for your patience. I am pleased to confirm that all the reviewers' concerns have been addressed. The response to Reviewer 5's comment on the non-independence modeling is satisfactory. The limitation has been described in a separate section of the Discussion, and I believe that future research can explore this issue further. Your manuscript is now generally ready for publication. Thank you for your thorough revisions in response to all the reviewers' feedback.